# Magnetic Levitation Patterns of Microfluidic-Generated Nanoparticle–Protein Complexes

**DOI:** 10.3390/nano12142376

**Published:** 2022-07-11

**Authors:** Luca Digiacomo, Erica Quagliarini, Benedetta Marmiroli, Barbara Sartori, Giordano Perini, Massimiliano Papi, Anna Laura Capriotti, Carmela Maria Montone, Andrea Cerrato, Giulio Caracciolo, Daniela Pozzi

**Affiliations:** 1NanoDelivery Lab, Department of Molecular Medicine, Sapienza University of Rome, Viale Regina Elena 291, 00161 Rome, Italy; luca.digiacomo@uniroma1.it (L.D.); erica.quagliarini@uniroma1.it (E.Q.); giulio.caracciolo@uniroma1.it (G.C.); 2Institute of Inorganic Chemistry, Graz University of Technology, Stremayrgasse 9/IV, 8010 Graz, Austria; benedetta.marmiroli@elettra.eu (B.M.); barbara.sartori@elettra.eu (B.S.); 3Dipartimento di Neuroscienze, Università Cattolica del Sacro Cuore, Largo Francesco Vito 1, 00168 Rome, Italy; giordano.perini@unicatt.it (G.P.); massimiliano.papi@unicatt.it (M.P.); 4Fondazione Policlinico Universitario A. Gemelli IRCSS, 00168 Rome, Italy; 5Department of Chemistry, Sapienza University of Rome, P.le A. Moro 5, 00185 Rome, Italy; annalaura.capriotti@uniroma1.it (A.L.C.); carmelamaria.montone@uniroma1.it (C.M.M.); andrea.cerrato@uniroma1.it (A.C.)

**Keywords:** magnetic levitation, protein corona, microfluidics, graphene oxide

## Abstract

Magnetic levitation (MagLev) has recently emerged as a powerful method to develop diagnostic technologies based on the exploitation of the nanoparticle (NP)–protein corona. However, experimental procedures improving the robustness, reproducibility, and accuracy of this technology are largely unexplored. To contribute to filling this gap, here, we investigated the effect of total flow rate (TFR) and flow rate ratio (FRR) on the MagLev patterns of microfluidic-generated graphene oxide (GO)–protein complexes using bulk mixing of GO and human plasma (HP) as a reference. Levitating and precipitating fractions of GO-HP samples were characterized in terms of atomic force microscopy (AFM), bicinchoninic acid assay (BCA), and one-dimensional sodium dodecyl sulfate–polyacrylamide gel electrophoresis (1D SDS-PAGE), and nanoliquid chromatography–tandem mass spectrometry (nano-LC-MS/MS). We identified combinations of TFR and FRR (e.g., TFR = 35 μL/min and FRR (GO:HP) = 9:1 or TFR = 3.5 μL/min and FRR (GO:HP) = 19:1), leading to MagLev patterns dominated by levitating and precipitating fractions with bulk-like features. Since a typical MagLev experiment for disease detection is based on a sequence of optimization, exploration, and validation steps, this implies that the optimization (e.g., searching for optimal NP:HP ratios) and exploration (e.g., searching for MagLev signatures) steps can be performed using samples generated by bulk mixing. When these steps are completed, the validation step, which involves using human specimens that are often available in limited amounts, can be made by highly reproducible microfluidic mixing without any ex novo optimization process. The relevance of developing diagnostic technologies based on MagLev of coronated nanomaterials is also discussed.

## 1. Introduction

Magnetic levitation is a reliable, portable, and simple technique that separates solids or water-immiscible organic liquids according to their densities [1]. Despite its early applications in the field of material science, MagLev is currently employed in nano- and biotechnological research, including studies on nanoparticles, biomolecules, and their mutual interactions. In this regard, MagLev can be employed to characterize the protein layer that adsorbs on nanoparticles (NPs) upon exposure to biological fluids, e.g., human plasma. Indeed, upon exposure to human plasma (HP), NPs are coated by a protein corona which is shaped by three categories of factors, i.e., the NP’s physical–chemical properties, the protein source, and environmental factors (e.g., shear stress and temperature). In 2014, some of us demonstrated that the NP–protein corona is personalized and disease-dependent [2,3,4]. This breakthrough paved the way for the development of nanoparticle-enabled blood (NEB) tests for disease detection. In the NEB test, NPs (e.g., liposomes, gold NPs, etc.) are incubated with HP from healthy subjects and cancer patients, leading to a library of personalized protein coronas. Corona proteins are isolated from NPs via centrifugation and characterized by using one-dimensional sodium dodecyl sulfate–polyacrylamide gel electrophoresis (1D SDS-PAGE) [5] and liquid chromatography–tandem mass spectrometry (LC-MS/MS) [6]. Statistical analysis of protein patterns allows for the distinguishing of cancer patients from healthy subjects with high sensitivity and specificity. Several variants of the NEB test have been developed for the detection of different cancer types such as pancreatic ductal adenocarcinoma (PDAC) [7], non-small cell lung cancer [8], and meningioma [9]. However, isolating proteins from NPs—the key step for running the test—can affect the reproducibility of experimental data. On the other hand, indirect characterization of the personalized protein corona (i.e., without isolating plasma proteins from NPs) may help towards reducing inter-user variability, thus allowing for comparison results of different laboratories. Among techniques allowing for indirect characterization of the protein corona, magnetic levitation (MagLev) has been recently employed to separate NP–protein complexes according to their densities [10]. Some of us demonstrated that the MagLev patterns of graphene oxide (GO)–protein complexes contain specific signatures of PDAC [11]. In a MagLev experiment, NP–protein complexes are formed by bulk mixing outside the magnetic field and then injected into the cuvette positioned inside the magnetic field. However, the recent literature points out that standardization of experimental protocols for each stage of the workflow is needed to increase the reliability of corona studies [12]. To this end, here, we used a microfluidic device for reproducible self-assembly of GO and plasma proteins and explored the effect of total flow rate (TFR) and flow rate ratio (FRR) on the MagLev patterns of microfluidic-generated graphene oxide–protein complexes. We identified a combination of microfluidic parameters leading to MagLev patterns with bulk-like features. As a typical MagLev experiment for disease detection is based on a sequence of optimization, exploration, and validation steps, this implies that the optimization (e.g., searching for optimal NP:HP ratios) and exploration (e.g., searching for MagLev signatures) steps can be performed using samples generated by bulk mixing. When these steps are completed, the validation step, which may involve using human specimens that are obtainable in limited amounts, can be performed without performing an ex novo optimization process.

## 2. Materials and Methods

### 2.1. Preparation of Graphene Oxide

Graphene oxide (GO) aqueous solution was procured by Graphenea (San Sebastián, Spain). GO solution (0.25 mg/mL) was subjected to sonication (Vibra cell sonicator VC505, Sonics, and Materials, Newton, CT, USA) to obtain homogenous GO sheets. Preliminary experiments were performed to evaluate the effect of GO size on the MagLev profiles. According to results reported in Appendix A and considerations thereof, we used GO sheets size of about 800 nm and negative zeta potential (i.e., −32 ± 2 mV). Further details of GO sizing, centrifugation, and characterization can be found elsewhere [13].

### 2.2. Preparation of GO-HP Samples

For all the experiments, commercial human plasma (HP) was purchased from Sigma-Aldrich, Inc. (Merk KGaA, Darmstadt, Germany). Lyophilized HP was dissolved in water according to the manufacturer’s instructions, then clarified by centrifugation, and finally stored at −20 °C. For the static mixing condition, GO (0.25 mg/mL) was incubated with HP at different GO:HP volume ratios (see Table 1 for further details). Distilled water was added to each sample until a total volume of 100 µL was reached. The desired total volume (100 μL), the component ratio (i.e., 9:1), and the incubation time were equal to those used in the microfluidic device. To obtain a static counterpart of a microfluidic mixing with total flow rate Q and total volume V, samples were incubated for a period of t = V/Q. In this work, Q was set equal to 3.5 μL/min, 7 μL/min, and 35 μL/min, while the total sample volume was fixed to 100 μL. Thus, the corresponding incubation times for the static mixing were 28.6 min, 14.3 min, and 2.86 min, respectively. Preliminary experiments were aimed at exploring whether the incubation time could affect the MagLev detection. Results reported in Appendix A showed that the incubation time had no appreciable effect on the MagLev profiles. For the microfluidic mixing, a commercial cross-shaped microfluidic device was used (Fluidic 394, microfluidic ChipShop GmbH, Jena, Germany), which featured squared channels of 200 µm width. The central channel length after the cross was 80 mm. Thus, the central channel volume was equal to 3.2 μL. The mixing was obtained through hydrodynamic focusing, a well-known technique to achieve fast mixing [14], according to which HP was inserted in the central channel and GO on the lateral ones, as shown in Figure 1a. The solutions were injected into the microfluidic device with three syringe pumps (Harvard Apparatus, Holliston, MA, USA). First, the ratio between the lateral flow rates and the central one was kept constant at 9:1, giving a central HP stream around 20 µm wide. Three different total flow rates were chosen, i.e., 3.5 μL/min, 7 μL/min, and 35 μL/min, to change the residence time of the solution inside the channels (i.e., residence time = channel volume/total flow rate, resulting in 55 s, 27.4 s, and 5.5 s, respectively) and therefore the mixing time. After selecting the most promising flow rate, two other ratios between lateral and central flow rates were investigated: 4:1 giving a stream 40 µm wide, and 19:1 to obtain 10 µm width, as reported in Table 1.

As a last step of the microfluidic mixing, samples were collected in a vial until a total volume of 100 μL was reached, thus undergoing an undesired but unavoidable static contact interaction. However, since protein adsorption on NPs usually occurs in a very short time period (about 0.5 min [15]), it can be reasonably supposed that GO–protein complexes that are formed during the dynamic incubation are subjected only to small modifications during their collection.

For both static and microfluidic mixing, GO-HP samples were prepared at a controlled temperature of 26 °C. Then, samples were inserted into the MagLev platform and observed for 20 min. Image processing analysis was automated and performed in about 5 min per sample. Once the image acquisition was finished, it took the cuvette cleaning time to switch from one sample to another (i.e., about 5 min). This step includes repeated washing with ethanol/water and conditioning with the paramagnetic medium used. As for the cleaning of the microfluidic device, ultrapure water was injected at high total flow rates into the microfluidic channels. Both plasma and graphene oxide were soluble in the aqueous solvent, so the cleaning of the channel and the sample collection tubes was guaranteed in about 10 min.

### 2.3. MagLev Device

The MagLev device consisted of two N42-grade neodymium (NdFeB) coaxial square permanent magnets (2.5 cm length, 2.5 cm width, and 5.0 cm height, purchased from Magnet4less), which face each other through N poles, with a separation distance of d = 2.8 cm. The strength of the magnetic field was ∼0.5 T on the surface of the magnet. The sample container was a plastic cuvette of 2 mL and 2.5 cm in height. As a paramagnetic solution, we used an aqueous solution of dysprosium (III) nitrate hydrate (salt purchased from Sigma-Aldrich, Inc., Merk KGaA, Darmstadt, Germany) concentrated at 80 mg/mL. Preliminary experiments for choosing the concentration of the paramagnetic solutions are reported in Appendix A.

Then, each GO-HP sample was injected at the bottom of the cuvette with a syringe and kept upright until the complete dissolution of the sample. When the whole sample volume lifted towards the surface and distributed homogenously, the cuvette was inserted between the magnets.

### 2.4. Fundamentals of MagLev

The MagLev technology can levitate a diamagnetic object in a paramagnetic solution when the magnetic and gravitational forces cancel out each other.
(1)$${\vec{F}}_{mag}+{\vec{F}}_{g}=0$$
where ${\vec{F}}_{mag}$ depends on the magnetic susceptibility of the paramagnetic medium ($\chi_{m}$), the magnetic susceptibility ($\chi_{s}$), the volume (*V*) of the diamagnetic object, the magnetic field ($\vec{B}$), and the magnetic permeability of free space ($\mu_{0}$) as follows:(2)$${\vec{F}}_{mag}=\frac{\chi_{s}-\chi_{m}}{\mu_{0}}V\left(\vec{B}\cdot \vec{\nabla }\right)\vec{B}$$
and ${\vec{F}}_{g}$ is the buoyancy-corrected gravitational force, which is
(3)$${\vec{F}}_{g}=\left(\rho_{s}-\rho_{m}\right)V\vec{g}$$
where the sample density and medium density are, respectively, represented by $\rho_{s}$ and $\rho_{m}$, and the gravity acceleration by $\vec{g}$. When the final equilibrium (Equation (1)) is reached, the diamagnetic sample reaches a steady height (*h*) that depends on the density of the object in the following paramagnetic solution:(4)$$h=\frac{d}{2}+\frac{\left(\rho_{s}-\rho_{m}\right)g\mu_{0}d^{2}}{\left(\chi_{s}-\chi_{m}\right)4B^{2}}$$

Equation (4) collects the information related to the expression of the magnetic force (Equation (2)), the gravitational force (Equation (3)), and the geometry of the magnetic setup. Finally, image series of MagLev patterns (at a controlled temperature of 26 °C) of both steady components and precipitating populations were acquired with a Nikon D5600 camera (time-lapse mode, 1 frame per 20 s) and processed using custom MATLAB (MathWorks, Natick, MA, USA) scripts. Briefly, for each frame, the vertical intensity profile was computed by averaging the recorded intensity over a region of interest containing the inner part of the cuvette. Then, after background subtraction, profiles were normalized to the maximum detected intensity over a reference window to avoid undesired effects due to exposure variations.

To evaluate the MagLev limit of detection and response, preliminary tests with protein samples and reference materials were performed. Results are reported in Appendix A.

### 2.5. Atomic Force Microscopy (AFM)

For AFM measurements, samples were prepared as explained elsewhere [16]. Briefly, a 20 µL aliquot of sample was deposited on sterile, freshly cleaved mica discs, air-dried, and measured with a NanoWizard II atomic force microscope (JPK Instruments AG, Berlin, Germany). The images were acquired using silicon cantilevers with high-aspect-ratio conical silicon tips (CSC37 MikroMasch, Tallinn, Estonia) characterized by an end radius of about 10 nm and a half conical angle of 20°. Cantilevers with a nominal spring constant of about k = 0.4 Nm^−1^ were thermally calibrated. Data analysis was performed via JPK instrument software.

### 2.6. Bicinchoninic Acid Assay (BCA)

For BCA measurements, we used a BCA Protein Assay reagent (Pierce, Thermo Scientific, Waltham, MA, USA) to measure the number of bound proteins on GO, according to the manufacturer’s protocol. GO-HP samples were centrifuged for 15 min at 4 °C, 21,400× *g*. Pellets were then washed three times with PBS to remove unbound and loosely bound proteins and finally obtain the so-called “hard corona.” Samples were resuspended in water, and then 10 µL of each sample was placed into a 96-multiwell plate, followed by adding 200 µL of BCA Protein Assay reagent. The multiwell was incubated at 37 °C for 30 min and then mixed on a plate shaker. The absorbance of each sample, blank, and standard was measured with a Glomax Discover System (Promega, Madison, WI, USA) at 560 nm. The protein concentration was calculated using the standard curve, and all the measures were made in triplicate.

### 2.7. Sodium Dodecyl Sulphate–Polyacrylamide Gel Electrophoresis (SDS-PAGE)

One-dimensional SDS-PAGE experiments were performed after protein corona isolation via centrifugation for 15 min at 4 °C, 21,400× *g*. Pellets were washed three times with PBS to remove unbound and loosely bound proteins and then resuspended in 20 mL of Laemmli Loading buffer 1× and boiled for 10 min at 100 °C. Each sample was loaded on a gradient polyacrylamide gel stain-free (4–20% TGX precast gels, Bio-Rad, Hercules, CA, USA) and run at 100 V for 150 min. Finally, gel images were acquired with a ChemiDoc™ gel imaging system (Bio-Rad, Hercules, CA, USA) and processed using custom MATLAB scripts (MathWorks, Natick, MA, USA), as previously reported [17].

### 2.8. Nanoliquid Chromatography–Tandem Mass Spectrometry

For nanoliquid chromatography–tandem mass spectrometry experiments, after the separation of proteins by SDS-PAGE, selected bands were excised from the gel and identified as previously reported [18]. In brief, the excised bands were reduced, alkylated, and then digested with trypsin. For each sample, 20 µL was analyzed. Briefly, a full scan and MS/MS analysis of eluting peptides were performed using an Orbitrap Elite hybrid ion trap–Orbitrap mass spectrometer (Thermo Scientific, Bremen, Germany) in the *m*/*z* range of 380–1400 Da and 30,000 (full-width at half maximum at *m*/*z* 400) resolution for the full scan and 15,000 resolution from MS/MS in top 10 data-dependent modes. For each sample, three technical replicates were performed, and the experiments were run twice. Spectra were analyzed using the open quantitative proteomics software MaxQuant (version v1.6.3.4) designed for analyzing large mass-spectrometric data sets [19].

## 3. Results and Discussion

In this work, we explored the effects of microfluidic parameters on the MagLev patterns of GO-HP samples. The experimental workflow is depicted in Figure 1. Briefly, GO was injected in the lateral inlets of the microfluidic device, while HP was fluxed in the central channel, both at controlled flow rates, for the desired incubation time (Figure 1a).

After the mixing, samples were collected at the end of the microfluidic cartridge, injected into the MagLev device (Figure 1b–d), and followed in time by acquiring image stacks (Figure 1e,f). Finally, by processing the image time series, the corresponding intensity profiles were computed (Figure 1g). As a first step, we studied the magnetic levitation of GO-HP samples for three different TFRs, i.e., 3.5 μL/min, 7 μL/min, and 35 μL/min. The total volume of the collected sample (100 μL) and the FRR (9:1 GO:HP) were not changed. Static incubation was carried out by incubating 90 μL with 10 μL HP in a test tube and used as a reference. Representative MagLev images corresponding to static and microfluidic mixing are shown in Figure 2a–d. They refer to the frame at which complete separation of the precipitating and levitating fraction occurred. As Figure 2e clearly shows, the abundance of the levitating fraction decreased monotonously with increasing TFR. At the highest TFR (i.e., 35 μL/min, blue curve), the MagLev patterns of GO-HP complexes obtained upon static incubation (grey curve) and microfluidic mixing were almost superimposable. As TFR dictates the sample residence time in the channels, these results indicate that larger TFR values lead to bulk-like behavior.

As a next step, we explored the effect of FRR on the MagLev patterns of GO-HP samples (Figure 3). For this purpose, we fixed the TFR to 3.5 μL/min and varied the GO:HP FRR, according to the values reported in Table 1. As the FRR affects the volume ratio of the mixing components in the collected samples, it was necessary to set different static references for each of the explored conditions (reported in Table 1).

Results displayed in Figure 3 show that for the minimum investigated flow ratio (i.e., 4:1, pink dashed line), a large precipitating fraction centered at about 11 mm dominated the levitating fraction centered at 17 mm. That profile was very similar to its static counterpart. On the other hand, at a flow ratio equal to 9:1 (solid red line), the MagLev profile exhibited two balanced fractions. For the static MagLev curve (solid grey line), a broadening of the precipitating peak was observed along with a significant decrease in the levitating population. Finally, at the highest explored flow rate ratio (i.e., 19:1), the precipitating fraction was barely detected, whereas the levitating peak represented the dominating portion of the profile. Under this condition, no clear differences between microfluidic (dotted purple line) and static (dotted grey line) incubation were observed.

From these results, one general conclusion could be drawn: Depending on TFR, FRR, and their combination, the microfluidic incubation can yield different or similar MagLev outputs concerning the static mixing in terms of relative abundance of the levitating and precipitating fractions. These findings demonstrate that one can use microfluidics to generate coronated NPs whose MagLev profiles will be superimposable to those of samples produced by bulk mixing. In a typical MagLev experiment based on a sequence of optimization, exploration, and validation steps [11,20], this implies that the optimization (e.g., searching for optimal NP:HP ratios) and exploration (e.g., looking for MagLev signatures) steps can be performed with coronated samples produced by bulk mixing. When these steps are completed, conclusions can be automatically extended to microfluidic incubation without the necessity of a new optimization process. Consequently, the use of microfluidics can be restricted to the validation step with obvious advantages in terms of saving time and experimental costs.

As the abundance of the levitating and precipitating fractions depended on the incubation protocol, we were prompted to investigate their composition. Levitating and precipitating fractions of GO-HP samples after magnetic levitation were characterized in terms of atomic force microscopy, bicinchoninic acid assay (BCA), 1D SDS-PAGE, and nanoliquid chromatography–tandem mass spectrometry experiments. AFM measurements revealed the presence of protein-coated GO sheets in both levitating and precipitating fractions (Figure 4a–c).

Protein aggregates can be recognized as bright spots with a large thickness (reaching up to 50–90 nm) on GO–protein layers, whose thickness ranged from 15 to 30 nm, suggesting that stacking of multiple sheets could have occurred. The adsorbed protein amount was measured by BCA (Appendix A), which revealed a larger protein amount on the precipitating fraction than on the levitating one. This outcome agrees with 1D SDS-PAGE experiments (Figure 4d, Appendix A). Indeed, the total lane intensity corresponding to the levitating samples was one order of magnitude smaller than the precipitating counterpart. One-dimensional SDS-PAGE measurements also provided the molecular weight distributions of those proteins populating the corona formed on GO (Figure 4e). Clear differences were detected, especially within the 60–80 kDa MW range. Protein contributions to the main peaks of 1D SDS-PAGE distributions are listed in Appendix A, as detected in mass spectrometry experiments.

Lastly, we explored the stability of microfluidic-generated GO-HP samples. As Appendix A shows, the MagLev profiles acquired immediately after the sample preparation and6 days later were very similar in terms of location and amplitude of levitating and precipitating peaks.

Finally, two more aspects of the experimental findings deserve attention. First, we characterized the protein corona: It cannot be excluded that other kinds of biomolecules (e.g., metabolites [21,22]) could migrate inside the cuvette. How and to what extent this could change the MagLev profile of coronated material will be the subject of future investigations. Lastly, we would like to comment on the choice of the paramagnetic medium. Both unbound and bound proteins may undergo denaturation in dysprosium (III) nitrate hydrate [23]. However, for diagnostic purposes, this would not represent a limitation, as the main aim of that approach would be the detection of global differences between healthy and cancer coronas, independently of the protein denaturation state.

## 4. Conclusions

This study explored the effect of microfluidic incubation on the MagLev profiles of NP-HP samples. We found combinations of TFR and FRR that produced microfluidic-generated samples whose MagLev signature is undisguisable from that of samples produced by bulk self-assembly. This may be relevant when the MagLev technique is used to separate human samples due to differences in their densities. Our findings imply that the optimization and exploration steps of the workflow can be carried out using samples generated by bulk mixing. The generation of samples via microfluidics can be limited to the validation step, which involves using human specimens that are often available in limited amounts. Globally, our results will stimulate future clinical validation studies that will combine the robustness of microfluidics with the discrimination ability of the MagLev technology.

## Figures and Tables

**Figure 1 nanomaterials-12-02376-f001:**
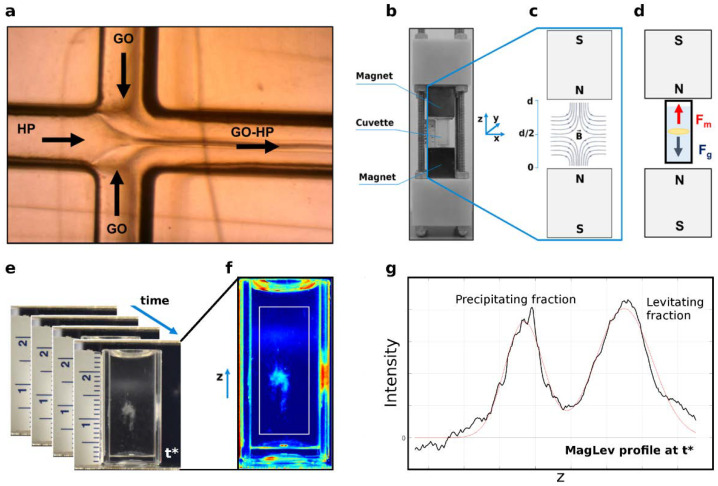
Representative workflow of the study: (**a**) GO-HP incubation within a microfluidic cartridge at controlled flow rates. Collected samples were inserted in (**b**) a MagLev device, responsible for a (**c**) magnetic field and thus a dynamic balance between (**d**) the acting forces, i.e., magnetic (F_m_) and gravitational (F_g_) force. Images of the samples were (**e**) acquired and (**f**) processed frame by frame to compute (**g**) the corresponding intensity profiles of the investigated samples (black lines). The profile was fitted by using a multipeak Gaussian distribution (red line). More details about image processing and the determination of the experimental error can be found in Appendix A.

**Figure 2 nanomaterials-12-02376-f002:**
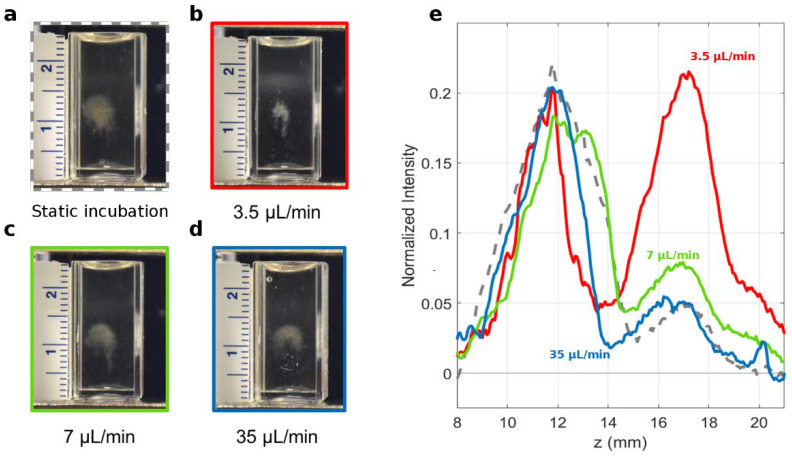
Representative MagLev patterns for GO-HP samples obtained via (**a**) static incubation and microfluidic incubation at (**b**–**d**) different flow rates. (**e**) Corresponding MagLev profiles of the investigated samples.

**Figure 3 nanomaterials-12-02376-f003:**
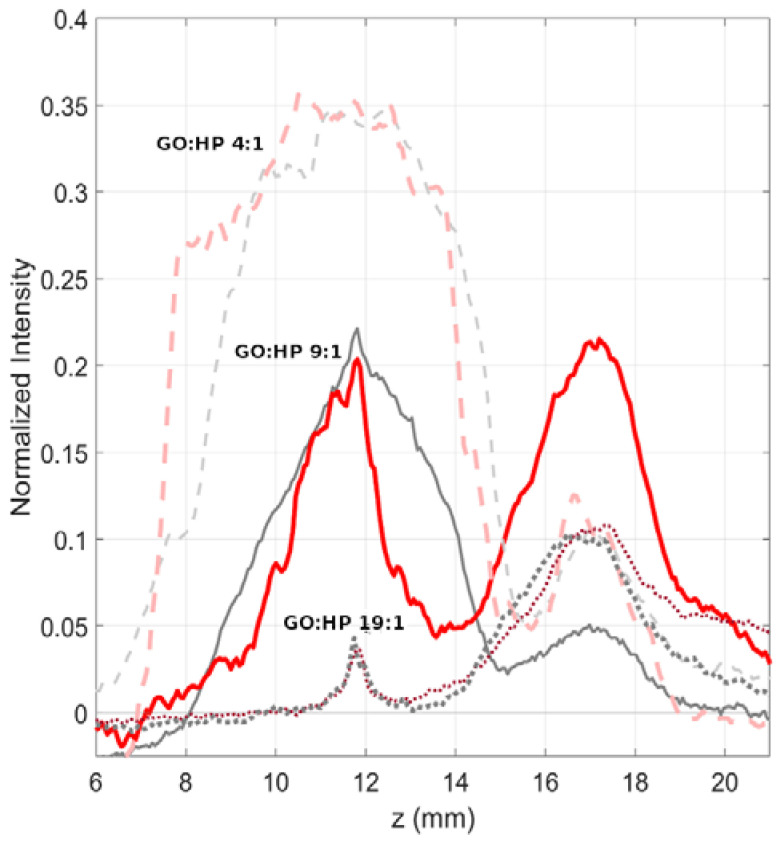
Representative MagLev patterns for GO-HP samples obtained via microfluidic incubation (red) with a TFR of 3.5 μL/min at different flow ratios: 4:1 dashed pink line, 9:1 solid red line, and 19:1 dotted purple line. Static incubation references for each condition are reported as dashed grey line, solid grey line, and dotted grey line, respectively.

**Figure 4 nanomaterials-12-02376-f004:**
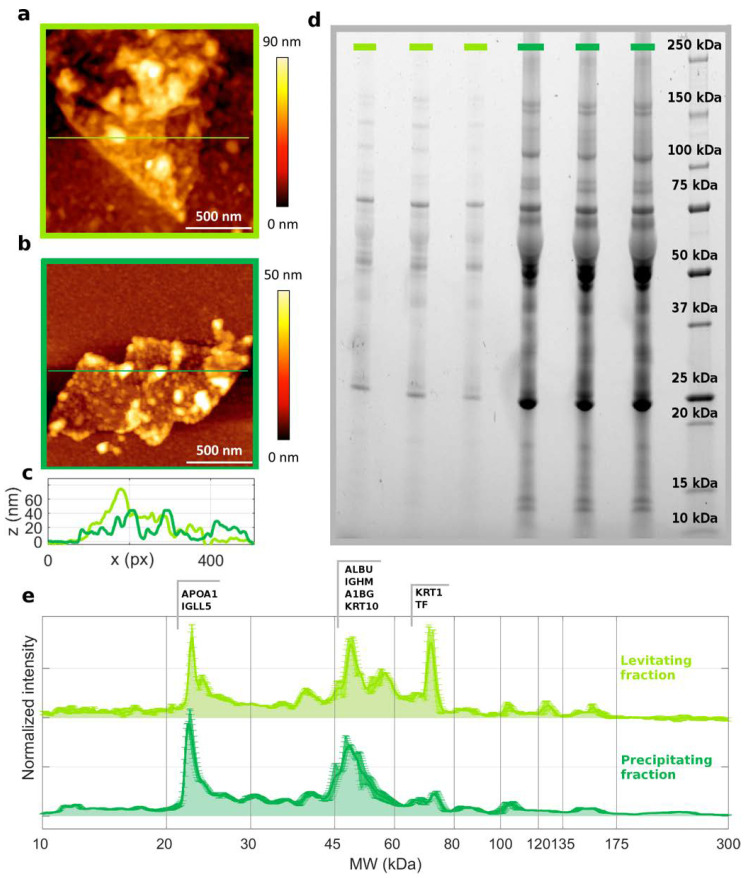
Atomic force microscopy (AFM) images for (**a**) levitating and (**b**) precipitating fractions of GO-HP samples collected after the MagLev measurements. (**c**) Z-profiles of the levitating (light green) and precipitating fractions (dark green). (**d**) Representative 1D SDS-PAGE image of the investigated samples (measurements were performed in triplicates) and (**e**) corresponding intensity profiles, which were normalized to the total lane intensity. The dominant protein contributions for the main peaks are reported in Appendix A, as detected in nanoliquid chromatography–tandem mass spectrometry experiments.

**Table 1 nanomaterials-12-02376-t001:** Microfluidic and static parameters employed to explore the effects of the FRRs on the MagLev patterns of GO-HP samples. The TFR was set at 3.5 μL/min. The GO flow rate corresponds to the sum of flow rates in the two lateral channels.

	Dynamic Incubation		Static Incubation	
GO:HP FRR	GO Flow Rate (μL/min)	HP Flow Rate (μL/min)	GO Amount (μL)	HP Amount (μL)
4:1	2.80	0.70	80	20
9:1	3.14	0.35	90	10
19:1	3.30	0.17	95	5

## Data Availability

The data presented in this study are available on request from the corresponding author.

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
