# Peer review of "Magnetic Levitation Patterns of Microfluidic-Generated Nanoparticle–Protein Complexes"

_nanomaterials, 2022, doi:10.3390/nano12142376_

Round 1
Reviewer 1 Report
In the authors’ work, the microfluidic device was used to prepared the nanoparticle-protein complexes (GO-HP), and the MagLev was used to separate the target protein according to their inherent density properties. This is a new detection technology that could be used in clinical with operability and repeatability. Some comments are shown below for improving the current work.
1. The detection accuracy of MagLev is related to many factors, such as the concentration of samples, amount of plasma proteins. One of the most impacting factors in the MagLev setup is the choice of the paramagnetic solution and its concentration, as the concentration affects both the solution’s density and magnetic susceptibility. However, none of these factors was evaluated in this manuscript.
2. The detailed description of GO-HP samples preparation was not clear enough in this manuscript. On line 90-91 “For the static mixing condition, GO (0.25 mg/mL) was incubated with HP for 1h at 37°C,” however, on line 108-109, “For both static and microfluidic mixing, GO-HP samples were prepared at a controlled temperature of 26 °C. GO-HP incubation was typically performed under static or dynamic conditions for a period ranging from 3-30 min.” What was the final incubation time and temperature? It is necessary to investigate the influence of incubation time and temperature on MagLev detection.
3. What was the concentration of the commercial human plasma, and the concentration of the final GO-PO sample. The concentration of HP will affect the determination of MagLev.
4. What is the final concentration and particle size of GO in the prepared GO-HP samples? It was necessary to characterize the GO nanoparticles. Would the size and zeta potential of nanoparticles affect the detection of MagLev?
5. Static incubation was used as control. An ordinary magnetic stirring condition could be added to see if only the microfluidic-generated nanoparticle-protein works best, while magnetic stirring was more convenient.
6. For the TFR investigation, on line 101-105, “Three different total flow rates were chosen, 3.5 μL/min, 7 μL/min, and 35 μl/min, to change the residence time of the solution inside the channels (respectively 5.5 s, 27.5 s, and 55 s) and therefore the mixing time after the beginning of the mixing.” Were the flow rates changing or were the residence time in the channels changing? How long did static incubation take which were as the control groups? In the microfluidic device (Figure 1), GO was injected through double channels while HP was injected through single channel. As being mentioned in Table 1, how to ensure the sum of the final total GO flow rate of the two channels is consistent with the settings?
Reviewer 2 Report
Authors have done intensive works on Magnetic Levitation patterns of microfluidic-generated nano- particle-protein complexes. The results are very comprehensive. However, the Magnetic Levitation patterns were mentioned at later text in abstract, introduction and methods. Please arrange the text to show Magnetic Levitation patterns in the earlier order of the text.
Magnetic levitation (MagLev) was mentioned in the first sentence of abstract. The next of Magnetic levitation was at line 30 and followed one is at line 61 in introduction.
This journal of nanomaterials, the audience will look for information of Magnetic Levitation patterns after see the title.
Material and method
'The MagLev device was purchased from Magnet4less' can be brought to material section.
Reviewer 3 Report
The article describes the optimization of several parameters to characterize GO-protein aggregates under magnetic levitation.
The article is well written and overall clear in describing the scientific results. However, I have a feeling the method description could be improved, essentially by adding a few sentences. I would suggest thus the following (very minor) revisions.
1) How is the sample transferred from its microfluidic channels into the 2ml cuvette? Is there a specific transfer procedure/pathway? Or is the solution from the microfluidic system just collected in a vial, then pipetted and mixed with the Dy3+ solution for magnetic levitation? if so, what about the time in the vial, that could count as static contact time.
2) Considering the image processing, it is a bit difficult to relate the picture to the intensity profile. One reason is that the diffuse band of the levitating aggregates is a bit difficult to notice compared to the clearly visible precipitate, yet has the same intensity. The horizontal white line in Figure 1f is also confusing, as I initially believed it was the integration path for the intensity. Maybe one way to be more clear would be to put both picture and intensity profile on the same z-axis, so features might be related.
Reviewer 4 Report
Authors should prepare more general introduction (see the reference Nanoparticles: From synthesis to applications and beyond, Advances in Colloid and Interface Science Volume 303, May 2022, 102640)
The authors should estimate the experimental error.
Is there any real discussion about the results?
What is the application in the medicine?
Round 2
Reviewer 4 Report
the paper can be accepted